# OpenReview forum: "Where to Begin? On the Impact of Pre-Training and Initialization in Federated Learning"
_ICLR.cc/2023/Conference — ICLR 2023 notable top 25%_

### Official Review · Reviewer_EYvh · 2022-10-24

**Confidence:** 3
**Correctness:** 3
**Technical Novelty And Significance:** 3
**Empirical Novelty And Significance:** Not applicable
**Recommendation:** 6

**Clarity, Quality, Novelty And Reproducibility:**

The paper is well organized but the presentation has minor details that could be improved.

The paper appears to be technically sound and the experimental results convincingly support the main claims.

The paper contributes some new ideas.

The author has submitted the code and the experimental details are sufficient. The contribution is clearly presented and the experimental results are thoroughly executed. I would tend to accept this paper.


**Strength And Weaknesses:**

Strength:

1)The paper is well written, and the experiments are sufficient.

2)The paper finds that starting federated learning from a pre-trained initialization reduces the effect of both data and system heterogeneity.

3)The experiments show that the pre-trained initialization leads to communication savings and reduces the overall training time in federated learning.

4)The findings in this paper suggest directions for future related work.

Weaknesses:

1)In Figure 1 and Figure 5, what is the representation of the curve radian?

2)The α in Figure 2 is not explained in the main paper.

3)Does the model pre-trained on public data introduce security issues to users? How do we make trade-offs between security and performance?


**Summary Of The Paper:**

This paper studies the impact of model initialization(random or pre-trained) on federated optimization methods. The experiments show that starting federated learning from a pre-trained initialization reduces the effect of both data and system heterogeneity. The experiments also show that the pre-trained initialization reduces the training time.

**Summary Of The Review:**

The paper is significant to model initialization(random or pre-trained) on federated optimization methods. The contribution is clearly presented and the experimental results are thoroughly executed. I would tend to accept this paper.

---

> ### Author Response · Authors · 2022-11-09
> **Response to reviewer EYvh**
>
> We thank the reviewer for their valuable suggestions and questions. We will revise the paper to include an explanation for \alpha in Figure 2, and discuss which application with public data available. In the meantime, we address the reviewer’s questions below.
>
> > In Figure 1 and Figure 5, what is the representation of the curve radian?
>
> The curvature has no meaning; it simply interpolates from the Pre-Trained and Random values. We chose this style to illustrate the change in ordering (lots of curves crossing).
>
> > The α in Figure 2 is not explained in the main paper.
>
> To generate the non-IID splits for CIFAR-10, we follow Hsu et al. (2019) and partition the dataset using a Dirichlet distribution with $\alpha = 0.1$. $\alpha$ dictates the degree of non-IIDness. As $\alpha \rightarrow \infty$, the data becomes more IID. We will add this detail to the revision.
>
> > Does the model pre-trained on public data introduce security issues to users? How do we make trade-offs between security and performance?
>
> We assume that the data used for pre-training is not subject to the same privacy concerns as data on the device. One can obtain these data from publicly available sources with an appropriate usage license or from users who opted in. In our work, our vision models were pre-trained on ImageNet, and our text models were pre-trained on wikitext-103.

---

> > ### Comment · Reviewer_EYvh · 2022-11-25
> > **Reply**
> >
> > Many thanks for the responses.
> > My concerns have been solved.

---

> ### Author Response · Authors · 2022-11-14
> **Revison**
>
> We uploaded the revised version of the paper with text highlighted in blue to show the changes. We would be happy to reply if the reviewer have any question.

---

### Official Review · Reviewer_Kusi · 2022-10-25

**Confidence:** 4
**Correctness:** 4
**Technical Novelty And Significance:** 3
**Empirical Novelty And Significance:** 3
**Recommendation:** 8

**Clarity, Quality, Novelty And Reproducibility:**

The article is well written, the observations are clearly stated and some of the limitations are highlighted. Code to reproduce the experiments is also provided

**Strength And Weaknesses:**

Strengths:
-The evaluations are extensive covering a wide array of tasks, datasets, and federated optimization settings
-The observations are potentially important: methods addressing heterogeneity may not be relevant in cases where pre-trained models can be used

No substantial weakness are noted

Some Comments:
- The effect of the pre-trained model is not heavily studied. It would be interesting to know if better pre-trained models lead to better results.
- I would like to see some discussion on how the learning rates selected are different (or same) in the pre-training/random setting
- It would be interesting to see the effect on more distant combinations of pre-training and target data. For example imagenet->medical/satellite image data
- In Chen et al it was shown that FedAVG can benefit even from pre-training on synthetic data. It would be interesting to see if such approaches (and the change in ranking of algos) hold for the algorithms evaluated here.




**Summary Of The Paper:**

The authors study federated learning starting from pre-trained models in both the image and text setting. They study a wide range of FL algorithms. Some key observations are that (a) the gap of iid to non-iid closes (b) ranking of FL algorithms is different under pre-training (c) effect of heterogneity are not severed. An investigation of how pre-training helps under heterogeneity is performed.


**Summary Of The Review:**

The paper has a number of important findings for guiding FL research and practice. The evaluations are thoroughly done across a wide range of datasets and tasks. Insights into the observed effects are also provided.

Post-Rebuttal: I have read the rebuttal and response to other authors. My review and score stand.

---

> ### Author Response · Authors · 2022-11-09
> **Response to reviewer Kusi**
>
> We thank the reviewer for their positive assessment of our work and the suggestions for improvements. In the next few days, we will update the rebuttal and revise the paper to include best learning rates. In the meantime, we address the reviewer's concerns below:
>
> > How are the learning rates selected in the pre-trained vs random setting?
>
> We will add the best learning rates in the Appendix in the revision. To select the best learning rates for each setting, we split the clients into train, eval, and test clients. We fixed server momentum for FedAvg and FedAdam $\beta_1 = 0.9$. We tune the client and server learning rates over the range [10^-6, 10] on the eval clients and report the final accuracies on the test clients.
>
> > Pre-training on synthetic data and different combinations of pre-training
>
> We agree that studying the effect of more distinct combinations of pre-training and target datasets or the pre-training on synthetic data is interesting. Given the time in the rebuttal period, we cannot perform these experiments.

---

> ### Author Response · Authors · 2022-11-14
> **Revision**
>
> We would like to check if the reader have read our rebuttal. We would be happy to reply if the reviewer have any question.

---

### Official Review · Reviewer_aMPP · 2022-10-25

**Confidence:** 4
**Clarity, Quality, Novelty And Reproducibility:** 1. The paper writing is clear and of …
**Correctness:** 3
**Technical Novelty And Significance:** 2
**Empirical Novelty And Significance:** 3
**Recommendation:** 8

**Strength And Weaknesses:**

## Strengths

**S1.** Thorough and extensive experimental evaluation of different optimization aspects which may be useful to the field of federated learning.

**S2.** Several drawn conclusions are novel and can be interesting for the community.

## Weaknesses

**W1.** The concept of using pre-trained weights (in federated deep learning) is not novel on its own. Some of the ideas were explored in previous works.

**W2.** Some important experimental details are missing. Namely, it is not clear how hyper-parameter tuning was performed. What clients/data were used for tuning step sizes, etc (the ranges also seem very limited)? It would be very helpful to show how pre-trained initialization affects the performance of the methods for different combinations of parameters (e.g., like it was done with heatmaps in the Adaptive FedOpt paper).  What kind of random initialization was used in the experiments? Prior literature showed that various strategies can lead to different results in the centralized case.

Please correct me, if I missed these details.

**W3.** The thesis
> Starting from a pre-trained model signiﬁcantly reduces the difference between having non IID vs IID data at clients.

is not well-supported (or inaccurately stated), in my view. It is not fully clear what is meant by IID data splits for naturally partitioned federated datasets in the paper. Was the data somehow reallocated between the same amount of clients or simply centralized training was performed? The latter case would require different wording in my view as it studies not the difference between non-IID and IID but rather federated and centralized cases. To properly show the effect of using the pre-trained model for a level of non IIDness one ideally needs to vary the level of heterogeneity for the same problem.

**W4.** Some of the Section 5 conclusions are questionable and doubtful. For example, the Gradient Diversity (GD) of client updates is lower for the pre-trained model, while the original paper by Yin et al., 2017 argues that better convergence is achieved for higher GD.  As far as I understand the plots show results of single runs which is not very reliable. Cosine similarity behavior looks quite random and similar for different initialization, apart from maybe the start. The connection to theory is not really convincing.

I would also like to ask how exactly the Hessian matrix was computed.

**Summary Of The Paper:**

The paper studies the effect of using pre-trained neural network weights on federated optimization. Several combinations of server and client optimizers are considered in combination with 2 different models for vision and language tasks. It was shown that pre-trained initialization reduces the negative effects of data and systems heterogeneity on training convergence and changes the ranking of optimization methods.

**Summary Of The Review:**

Overall, the paper represents a piece of good-quality research. I think that there are enough insights for publication. There is also a potential for an increase in my score if some of the questions and concerns are addressed by the authors.

---

> ### Author Response · Authors · 2022-11-09
> **Response to reviewer aMPP (1/3)**
>
> We thank the reviewer for their constructive and valuable feedback! In the next few days, we will update the rebuttal and revise the paper to include details about the learning rates, the heatmap for the hyper-parameters, the IID partitioning process, and how the Hessian matrix was computed. In the meantime, we discuss the various points and questions raised by the reviewer below.

---

> > ### Author Response · Authors · 2022-11-09
> > **Response to reviewer aMPP (2/3): Addressing W1 to W4**
> >
> > > W1: The concept of using pre-trained weights (in federated deep learning) is not novel on its own.
> >
> > We agree that the concept of pretraining is not novel. However, our contribution is not proposing the use of pretraining for FL but rather the analysis of the importance and effects of pretraining in FL. Moreover, we show that starting from pre-trained weights can alter the choice of which FL algorithms may be preferred or best performing, and we offer recommendations on how to evaluate new FL algorithms.
> >
> > > W2: Some important experimental details are missing.
> >
> > What clients/data were used for tuning step sizes?
> > For each dataset, we split the clients into train clients, eval clients, and test clients. We fixed server momentum for FedAvg and FedAdam $\beta_1$ at 0.9. We tune the client and server learning rates over the range [10^-6, 10] on the eval clients and report the final accuracies on the test clients.
> >
> > It would be very helpful to show how pre-trained initialization affects the performance of the methods for different combinations of parameters (e.g., like it was done with heatmaps in the Adaptive FedOpt paper).
> > Thank you for the suggestion. We are working on producing heatmaps for client and server learning rates and will add them in the revision. We’ll follow up here once they’re ready.
> >
> > > What kind of random initialization was used in the experiments?
> >
> > For all random initialization experiments, we use the default initialization schemes for each model. We reference the random initialization code used in our experiments below.
> > * Resnet18: https://github.com/pytorch/vision/blob/main/torchvision/models/resnet.py#L208:L214
> > * Squeezenet: https://github.com/pytorch/vision/blob/main/torchvision/models/squeezenet.py#L85:L93
> > * DistilGPT2: https://github.com/huggingface/transformers/blob/main/src/transformers/models/gpt2/modeling_gpt2.py#L454:L479
> > * CharLSTM: https://github.com/seongjunyun/Character-Aware-Neural-Language-Models/blob/master/model.py#L71:L85
> >
> > > W3 The study between non-IID and IID
> >
> > We added the process for IID partition in the revision. For IID, we pooled the entire dataset together, shuffled the data, and then partitioned it into the same number of clients as our non-IID partition (the former case in your question), the same process as McMahan et al., 2017.
> >
> > We show the result for varying levels of non-IIDness on CIFAR-10 below. We follow Hsu et al. (2019) to generate the different splits and partition the dataset using a Dirichlet distribution with $\alpha \in [0.1, 1.0, 10]$. As $\alpha \rightarrow \infty$, the data becomes more IID. We train a Squeezenet model for 1000 rounds using FedAdam at the server and SGD at the client.
> >
> > | Alpha | Pre-trained | Random |
> > |-------|-------------|--------|
> > | 0.1   | 75.9        | 50.6   |
> > | 1.0   | 78.1        | 66.8   |
> > | 10    | 80.1        | 72.8   |
> >
> > *  McMahan, Brendan, et al. "Communication-efficient learning of deep networks from decentralized data." Artificial intelligence and statistics. PMLR, 2017
> > *  Tzu-Ming Harry Hsu, Hang Qi, and Matthew Brown. Measuring the effects of non-identical data distribution for federated visual classification. arXiv preprint arXiv:1909.06335, 2019
> >
> > > W4. Some of the Section 5 conclusions are questionable and doubtful.
> >
> > > Gradient Diversity and Cosine Similarity
> >
> > Yin et al. argues that larger gradient diversity (GD) means that it’s possible to increase the batch size and use larger learning rate when training on IID samples without any local updates. In our work, we focus on the federated setting on non-IID data. Moreover, we use GD here as a metric to diagnose the training process.
> >
> > > I would also like to ask how exactly the Hessian matrix was computed.
> >
> > Thank you for clarifying the question. To compute the Hessian matrix, We also examine the largest eigenvalue of the Hessian matrix at initialization, round 0, using Power Iteration from PyHessian by Yao et., al 2020. We will add the citation to the paper and will include a discussion of these details in the revision.
> >
> > * Yao, A. Gholami, K Keutzer, M. Mahoney. PyHessian: Neural Networks Through the Lens of the Hessian, Spotlight at ICML workshop on Beyond First-Order Optimization Methods in Machine Learning, 2020

---

> > > ### Author Response · Authors · 2022-11-09
> > > **Response to reviewer aMPP (3/3): Clarity, Quality, Novelty and Reproducibility**
> > >
> > > > For instance, why did the previous papers not find that pre-training does not help to tackle heterogeneity issues?
> > >
> > > To our knowledge, very few papers investigate the impact of pre-training on alleviating heterogeneity. Our work is the first to systematically study both forms of heterogeneity, data-induced, and system-induced, across visual and language tasks on 15 SOTA federated optimization algorithms. We offer theoretical and empirical explanations for why pre-training is beneficial to FL.
> > >
> > > > I would like to ask why these particular deep learning models were chosen.
> > >
> > > We chose these models because they are (1) small enough to fit on-device, (2) have publicly available pre-trained weights, and (3) have different architectures and levels of capacity.
> > >
> > > * Squeezenet is a CNN-based model that can be compressed to less than 0.5 MB and thus can be fit on many mobile phones.
> > > * ResNet18 is widely used in previous FL literature (Hsieh et., al 2019; Reddi et., 2020).
> > > * CharLSTM is analogous to a simple LSTM, with the word embedding replaced with character embedding followed by a CNN layer to reduce the model size. This architecture is commonly used in production on-device applications (Hard et al. 2018).
> > >
> > > To our knowledge, DistilGPT2 is the smallest transformer in Hugging Face with a casual LM head.
> > >
> > >
> > > * Kevin Hsieh, Amar Phanishayee, Onur Mutlu, and Phillip B Gibbons. The non-IID data quagmire of decentralized machine learning. arXiv preprint arXiv:1910.00189, 2019.
> > >
> > > * Sashank Reddi, Zachary Charles, Manzil Zaheer, Zachary Garrett, Keith Rush, Jakub Konecnˇ y, ` Sanjiv Kumar, and H Brendan McMahan. Adaptive federated optimization. arXiv preprint arXiv:2003.00295, 2020.
> > >
> > > * Hard, Andrew, et al. "Federated learning for mobile keyboard prediction." arXiv preprint arXiv:1811.03604 (2018).
> > >
> > > > It would be helpful to briefly and explicitly mention the client updates formulas in the main text or Appendix.
> > >
> > > Thank you for the suggestion, we will add the client update formulas to the Appendix.

---

> ### Author Response · Authors · 2022-11-14
> **Revision**
>
> We uploaded the revised version of the paper with text highlighted in blue to show the changes. We would be happy to reply if the reviewer have any question. We are working to get the heat map of the hyper-parameters. We hope to get those numbers in a few days.

---

> ### Author Response · Authors · 2022-12-03
> **Clarification**
>
> We thank the reviewer for the positive feedback on our paper. Since the discussion period is coming to an end, we would love to hear back from you if you have any other questions.

---

> ### Comment · Reviewer_aMPP · 2022-12-14
> **Post rebuttal**
>
> I would like to thank the authors for their excellent work. All of my main complaints were resolved, or a promise was made to do so in the camera-ready. That is why I increased the score and can confidently recommend acceptance of this paper.
>
> It would be great if heatmaps for different hyperparameters for “natural” federated datasets (like FEMNIST and Stackoverflow) and more reasonable values of the stepsizes (for the current choice, most of the results are very poor, e.g., around 10% for CIFAR-10) are included in the camera-ready version.
>
> Some of my more minor concerns regarding Section 5 were not addressed (regarding multiple runs and connection to theory), but I hope the authors will take them into account for the camera-ready.

---

### Author Response · Authors · 2022-11-18
**Summary of changes**

We thank all reviewers for their feedback. We believe our replies address the questions raised in the reviews. We are happy to address any other questions in the remaining time.

Following suggestions from the reviewers, we have the following changes:
1. Added the learning rates for Figure 1
2. Outlined how we generate the IID partitions
3. Include CIFAR-10 with varying levels of heterogeneity.

We are unable to produce the heatmap with a grid of hyper-parameters before the end of the rebuttal period but we would be able to include this in the camera ready version.

---

> ### Author Response · Authors · 2022-11-22
> **Heatmap of Hyper-parameters**
>
> We show the combinations of server learning rate (ηg ) and client learning rate (ηg ) on
> CIFAR-10 with SGD at the client and various server optimizers. Overall, Figure 7 shows that pre-
> training requires to smaller client learning rates, roughly one order of magnitude smaller compared
> to random initialization.
>
> https://ibb.co/hXSh7dh

---

### Public Comment · ~Yue_Tan2 · 2023-02-02
**Relation of This Paper to Prior Work**

Dear Authors,

Congratulations on your well written paper which is accepted as the notable-top-25% in ICLR 2023. Great job!

Here, I would like to bring your attention to one of our prior works that also explores pre-training in FL. Your paper aims to study the impact of pre-training from the perspective of initialization and how it alleviates the heterogeneous issues, while ours mainly focuses on integrating pre-trained models into FL and exploring how to effectively learn from pre-trained models in FL [1].

I think that "pre-training for FL" is quite an interesting and promising direction. Would you mind discussing this in your camera-ready version?

Thank you.

Yue Tan

[1] Federated Learning from Pre-Trained Models: A Contrastive Learning Approach. NeurIPS 2022.

---

> ### Author Response · Authors · 2023-02-23
> **Thank you for the reference**
>
> Thank you for the reference, we will discuss your work in the camera ready version

---

### Decision · Program_Chairs · 2023-01-20

**Decision:**

Accept: notable-top-25%

**Justification For Why Not Higher Score:**

The paper has very interesting empirical findings. It could have been an oral if the authors were able to demonstrate the reasons behind their observations.

**Justification For Why Not Lower Score:**

It's a good paper with high scores.

**Metareview: Summary, Strengths And Weaknesses:**

The paper studies the impact of initializing from a pre-trained model (which is trained using some other/available data set) on the overall performance of the model (after being trained using the actual data). The paper provides a number of interesting messages through extensive numerical studies: 1) Pre-trainig helps to achieve a better performance. 2) Pretraining helps to alleviate the  effect of both data and system heterogeneity.

All the reviewers found the results very useful and recommended accepting the paper.

I would like to add one point (given the last two sentences of the abstract): If we look at data heterogeneity from the lens of representation theory, then the results of the paper would make a lot of sense. There is a line of work on FL that argues that in many FL datasets and applications, even though the users data are heterogenous, they often have a shared representation. And this shared representation can be exploited to learn better models. In the case of pertaining using another (often rich) dataset, a good representation will already be learned through the pertained model. In other words, the pertained model will provide a good representation of the data that can be further adapted  ti the data of the users in the main FL training procedure. I recommend that the authors look at the literature of defeated representation learning and establish a connection to their observations in the related works. A few papers that come to my mind are the following:

1. Collins et al: Exploiting Shared Representations for Personalized Federated Learning
2. Pillutla et al: Federated learning with partial model personalization
3. Collins et al: FedAvg with Fine Tuning: Local Updates Lead to Representation Learning

**Note From Pc:**

if the above contains the word "oral" or "spotlight" please see: "oral" presentation means -> notable-top-5% and "spotlight" means -> notable-top-25%. As stated in our emails, we are disassociating presentation type from AC recommendations